

# Orphan nuclear receptor NR4A2 inhibits hepatic stellate cell proliferation through MAPK pathway in liver fibrosis

Pengguo Chen[1,2], Jie Li[1], Yan Huo[1], Jin Lu[1], Lili Wan[1], Bin Li[1], Run Gan[1] and Cheng Guo[1,2]

[1] Department of Pharmacy, Shanghai Jiao Tong University Affiliated Sixth People's Hospital, Shanghai, China
[2] Shanghai Jiao Tong University School of Medicine, Shanghai, China

## ABSTRACT

Hepatic stellate cells (HSCs) play a crucial role in liver fibrosis, which is a pathological process characterized by extracellular matrix accumulation. NR4A2 is a nuclear receptor belonging to the NR4A subfamily and vital in regulating cell growth, metabolism, inflammation and other biological functions. However, its role in HSCs is unclear. We analyzed NR4A2 expression in fibrotic liver and stimulated HSCs compared with control group and studied the influence on cell proliferation, cell cycle, cell apoptosis and MAPK pathway after NR4A2 knockdown. NR4A2 expression was examined by real-time polymerase chain reaction, Western blotting, immunohistochemistry and immunofluorescence analyses. NR4A2 expression was significantly lower in fibrotic liver tissues and PDGF BB or TGF-$\beta$ stimulated HSCs compared with control group. After NR4A2 knockdown $\alpha$-smooth muscle actin and Col1 expression increased. In addition, NR4A2 silencing led to the promotion of cell proliferation, increase of cell percentage in S phase and reduced phosphorylation of ERK1/2, P38 and JNK in HSCs. These results indicate that NR4A2 can inhibit HSC proliferation through MAPK pathway and decrease extracellular matrix in liver fibrogenesis. NR4A2 may be a promising therapeutic target for liver fibrosis.

Corresponding author
Cheng Guo, guopharm@126.com

## INTRODUCTION

Liver fibrosis is a pathological process characterized by accumulation of extracellular matrix. It may be triggered by inflammation, drug, alcohol, virus and cholestasis (*Cassiman et al., 2002*). A variety of cells such as endothelial cells, hepatic stellate cells (HSCs) and Kupffer cells are involved in liver fibrogenesis. Among them HSCs play a crucial role and are the main cell source of extracellular matrix (*Reynaert et al., 2002*). HSCs, also known as Ito cells or perisinusoidal cells, are pericytes found in the perisinusoidal space of the liver. When the liver is damaged, quiescent HSCs are activated.

MAPK/extracellular regulated kinase (ERK) pathway is vital for the activation of HSCs. Recently liver fibrosis was found to be reversible which was demonstrated in animal models (*Cao et al., 2004*; *Saxena et al., 2007*). However, some mechanisms are still unknown.

There are increasing studies on NR4A subfamily. NR4A members are widely distributed in cells mediating differentiation, proliferation and apoptosis and are involved in many diseases such as cancer, vascular sclerosis and metabolic syndrome. NR4A members share highly conserved cellular region: a variable amino terminal area, a DNA-binding area at the center and a variable and continuous D-area that connects DBD to carboxyl terminal conservative E/F area. NR4A members are early phase reaction genes and serve as transcription factors (*Rius et al., 2006*; *Zhao, Desai & Zeng, 2011*; *Holla et al., 2011*). *Palumbo-Zerr et al. (2015)* observed that NR4A1 recruits a repressor complex limiting profibrotic TGF-$\beta$ effects. Persistent activation of TGF-$\beta$ suppresses NR4A1 expression in fibrotic diseases. *Yin et al. (2013)* discovered that expression of NA4A1, NR4A2 and NR4A3 decrease apparently in uterine fibroids. Studies on NR4A in liver disease are scarce. It is well confirmed that downregulation of ERK1 in HSCs remarkably attenuated the extracellular matrix deposition in fibrotic liver (*Zhong et al., 2009*). ERK5/MAPK in combination with NR4A2 results in boosting of transcriptional activity (*Sacchetti et al., 2006*). In addition, NR4A2 is identified as the target of ERK2 (*Zhang et al., 2007*). Taken together, NR4A2 may modulate HSCs in liver fibrosis.

In the present study we confirmed NR4A2 expression decreased significantly in fibrotic liver tissues and PDGF BB or TGF-$\beta$ stimulated HSCs. NR4A2 silencing led to higher expression of $\alpha$-smooth muscle actin, promotion of cell proliferation and increase of cell percentage in S phase as well as reduced phosphorylation of ERK1/2, P38 and JNK in HSCs. Thus NR4A2 may be a potential target for anti-fibrotic treatment.

## MATERIALS & METHODS

### Antibodies and reagents

Antibodies against glyceral-dehydephosphate dehydrogenase (GAPDH) (SC-25778), NR4A1 (SC-5569), NR4A2 (SC-991), NR4A3 (SC-30154) and $\alpha$-SMA (SC-32251) were purchased from Santa Cruz (CA, USA). Alexa Fluor® 546 IgG antibody (A10040) and Alexa Fluor® 488 IgG (H + L) antibody (A21202) were obtained from Life Technologies (Carlsbad, CA, USA). IRDye680 anti-rabbit antibody (926-32221) and IRDye680 anti-mouse antibody (926-32220) were from LI-COR Biosciences (Lincoln, NE, USA). Antibody against Erk1/2 (4695P), P-erk1/2 (4370P), JNK (9258P), P-JNK (4668P), P-P38 (4511P) and P38 (8690P) were purchased from Cell Signaling (Danvers, MA, USA). Nycodenz (QK1002424-1) was from Nycomed (Zurich, Switzerland). NC nitrocellulose was from Pall (Port Washington, NY, USA). Deoxyribonuclease I was obtained from Roche (Basel, Switzerland). D-HanK's and 30% acrylamide solution were from Bio-Light (Shanghai, China). Diaminobenzidine (DAB) Kit (GK50705) was from GeneTech (Shanghai, China). Cell cycle and Apoptosis Analysis Kit were purchased from R&S (Shanghai, China). Annexin V-FITC Apoptosis Detection Kit was from Keygen (NJ, China). TGF-$\beta$1 was obtained from Peprotech (Rocky Hill, NJ, USA). Tween-20 and ammonium persulfate were obtained from Sinopharm (Beijing, China). Lipofect-amineTM 2000 was purchased from Invitrogen (Carlsbad, CA, USA). Collagenase IV, PDGF BB and Pronase were purchased from Sigma (St. Louis, MO, USA). Fecal calf serum (FBS) was purchased

from Gibco (Waltham, MA, USA). RIPA buffer, phenylmethylsulfonyl fluoride, TEMED, 5× SDS loading buffer and BCA kit were from Beyotime (Shanghai, China).

## Animals and liver samples

Male Sprague-Dawley rats (SLAC, Shanghai, China), weighing approximately 250–270 g were maintained in a 12-h dark/12-h light cycle. All animal experiments were approved by the Animal Care and Use Committee of Shanghai Jiao Tong University Affiliated Sixth People's Hospital (License No. SYXK2011-0128). Human liver samples were offered by hepatobiliary surgery department of Shanghai Jiao Tong University Affiliated Sixth People's Hospital. All samples were obtained in accordance with the ethics committee of Shanghai Jiao Tong University.

## Cell culture

The HSC-T6 cells were kindly provided by Dr. Friedman of Mount Sinai School of Medicine of New York University (MSSM). Cells were cultured in DMEM medium supplemented with 10% heat-inactivated fetal calf serum (FBS). All cells were maintained in a humidified incubator at 37 °C with 5% $CO_2$. Platelet-derived growth factor (PDGF) BB or TGF-$\beta$ was used for stimulation.

## RNA isolation and quantitative real time-polymerase chain reaction

Total RNA was extracted from HSC-T6 cell using Trizol (Takara, Dalian, China) and reverse transcribed using PrimeScript RT Master Mix (Takara). Real time polymerase chain reaction (RT-PCR) was performed using SYBR Green PCR Kit (Takara) and Applied Biosystems 7500 real-time PCR system (Applied Biosystems, Foster City, CA, USA). Primer sequences (Sangon Biotech, Shanghai, China) were as follows: NR4A1 (Fw: 5′-ACACCGGAGAGTTTGACACC-3′; Rev: 5′-GGGTAGCAGCCATACACCTG-3′); NR4A2 (Fw: 5′-AGATTCCTGGCTTTGCTGAC-3′; Rev: 5′-CTGGGTTGGACCTGTATGCT-3′); NR4A3 (Fw: 5′-GGCTGCAAGGGCTTCTTCA-3′; Rev: 5′-CACCATCCCGACACTGAGA CA); $\alpha$-SMA (Fw: 5′-CCAGGGAGTGATGGTTGGA-3′; Rev: 5′-CCGTTAGCAAGGTCG GATG-3′); Col1 (Fw: 5′-AGGCATAAAGGGTCATCGTG-3′; Rev: 5′-ACCGTTGAGTCC ATCTTTGC-3′); $\beta$-actin (Fw: 5′-ACCCACACTGTGCCCATCTATG-3′; Rev: 5′-AGAGTACTTGCGCTCAGGAGGA-3′). The cycle conditions were 95 °C for 30 s, 40 cycles at 95 °C for 5 s and 60 °C for 34 s, one cycle at 95 °C for 15 s, 60 °C for 1 min and 95 °C for 15 s. The resulting sequences were normalized to the $\beta$-actin expression levels, and relative gene expression was measured by the $2^{-\Delta\Delta CT}$ method.

## Immunohistochemistry and immunofluorescence

Hepatic tissues were embedded in paraffin and sectioned. The sections were incubated in 3% $H_2O_2$ in methanol and nonspecific binding was blocked with 10% normal goat serum. The sections were incubated with primary antibody at 4 °C overnight, washed and incubated with secondary antibody for 60 min at room temperature. Antigen–antibody complexes were visualized using DAB kits.

Immunofluorescence was performed as described previously (*García-Pérez et al., 2013*; *García-Pérez et al., 2015*). For detecting NR4A2 and $\alpha$-smooth muscle actin ($\alpha$-SMA) expression in liver tissue, the sections were incubated for 48 h at 4 °C with the

following primary antibodies: NR4A2 (1:50), Alexa Fluor® 546 IgG (1:500), Alexa Fluor® 488 IgG (H + L) (1:500). The sections were incubated in 4, 6-diamino-2-phenylindole (DAPI, 1:100) for 1 min and mounted in antifade reagent.

## NR4A2 siRNA transfection

For RNA interference assay, the cells were seeded into 6-well cell culture plates at a density of $5 \times 10^5$ cells. The cells were transfected with siRNAs and Lipofectamine 2000 according to the manufacturer's instructions. The double strands of small interfering RNAs (siRNAs) targeting NR4A2 (GenePharma, Shanghai, China) were as follows: siRNA (1) (Fw: 5′-CGCGAAAUAUGUGUGUUUATT-3′; Rev: 5′-UAAACACACAUAUUUCGCGTT-3′); siRNA (2) (Fw: 5′-GACCAUGUGACUUUCAAUATT-3′; Rev: 5′-UAUUGAAAGUCACA UGGUCTT-3′); siRNA (3) (Fw: 5′-GACCUCACCAACACUGAAATT-3′; Rev: 5′-UUUCA GUGUUGGUGAGGUCTT-3′); Negative control (Fw: 5′-UUCUCCGAACGUGUCACG UTT-3′; Rev: 5′-ACGUGACACGUUCGGAGAATT-3′). Briefly, 100 nmol siRNAs and 5 µl Lipofectamine 2000 were diluted in serum and antibiotic free opti-DMEM (Invitrogen, Carlsbad, CA, USA) at a final volume of 500 µl. After mixing for 20 min at room temperature, the siRNA/Lipofectamine 2000 mixture was added to the cells and incubated at 37 °C in a humidified $CO_2$ incubator. Nonsilencing siRNA with no known homology to rat genes was synthesized as a negative control.

## Primary HSC isolation

Sprague-Dawley rats (SLAC, Shanghai, China) for primary HSCs were prepared. After anesthesia, the rat abdominal wall was opened and the intestines were moved aside so as to expose the vena cava and portal vein. A fine cannula ligated and fixed to a proper location was inserted into the vein after placing a thread around it. Perfusion solutions were incubated at 37 °C before usage. An adjustable peristaltic pump was then used to control the *in situ* liver perfusion process. After the liver swelled, the cannula in the inferior vena cava was opened to allow draining. Subsequently, the liver was perfused in turn with D-HanK's solution, Collagenase IV solution, Pronase solution for 15 min at 37 °C. The cell fraction was collected, washed and put between a top layer of buffer and a bottom cushion of 18% Nycodenz. After centrifugation at 1,400 g for 22 min, the HSCs fraction at the interface between the top and intermediate layer was obtained. Finally, the cells were cultured in DMEM medium supplemented with 10% FBS.

## Western blotting

Western blot analysis was performed according to standard protocols as described previously (*Ogawa et al., 2012*). Western blots were performed using antibodies against $\alpha$-SMA (1:500), GAPDH (1:1,000), NR4A1 (1:500), NR4A2 (1:500), NR4A3 (1:500), IRDye680 anti-rabbit (1:5,000), IRDye680 anti-mouse (1:5,000), Erk1/2 (1:5,000), P-erk1/2 (1:1,000), JNK (1:1,000), P-JNK (1:1,000), P-P38 (1:1,000) and P38 (1:1,000).

Protein samples were subjected to SDS-PAGE and then transferred onto a polyvinylidenefluoride membrane. After blocking, the membranes were incubated with primary antibodies followed by peroxidase-conjugated secondary antibodies. The protein samples were scanned through Odyssey Infrared Imaging System.

### CCK-8 assay

Cell viability was assessed using the Cell Counting Kit-8 (Dojindo, KMJ, Japan) according to the manufacturer's instructions. Briefly, HSC-T6 cells transfected with NR4A2 siRNA were seeded into 96-well plates ($3 \times 10^3$ cells per well). Next, 10 μl of CCK8 was added to each well and the cells were incubated for 2 h at 37 °C. Absorbance was assessed at 450 nm with a microplate reader (Bio-Tek, Winooski, VT, USA). The mean optical density (OD) of the wells in each group was used to draw cell growth curve.

### Cell cycle analysis

Cells were seeded into 6-well cell culture plates at a density of $5 \times 10^5$ cells. The cells were transfected with siRNA NR4A2 and 48 h later the cells were harvested, stained and fixed with propidium iodide (PI) according to the manufacturer's protocol and subject to cell cycle analysis using a flow cytometer (BD, Franklin Lakes, NJ, USA).

### Cell apoptotic assay

Cells were seeded into 6-well cell culture plates at a density of $5 \times 10^5$ cells. The cells were transfected with siRNA NR4A2 and 48 h later the cells were harvested. Apoptosis assay was measured using FITC AnnexinV Apoptosis Detection Kit according to the manufacturer's protocol and subject to cell cycle analysis using flow cytometer.

### Statistical analysis

The results were expressed as mean ± SD and were analyzed by analysis of variance (ANOVA) or Student's $t$-test followed by paired comparison as appropriate. $P < 0.05$ was taken as the minimum level of significance.

## RESULTS

### NR4A2 expression was reduced in activated HSCs

HSCs play an important role in liver fibrosis. Quiet HSCs could be activated by numerous factors and produce extracellular matrix. $\alpha$-SMA is a well-established marker for liver fibrosis. PDGF and TGF-$\beta$ are strong stimulating factors. After HSC-T6 cells was stimulated by PDGF BB (10 ng/ml or 20 ng/ml), NR4A2 mRNA level decreased significantly whereas $\alpha$-SMA level increased significantly (Fig. 1A). The level of NR4A2 protein has significantly reduced after treatment with PDGF BB (10 ng/ml, 20 ng/ml respectively) in HSC-T6 cells, yet inconspicuous changes occurred in NR4A1 and NR4A3 expression (Fig. 1B). Similar observation has been made after treatment with TGF-$\beta$ (2 ng/ml) (Fig. 1C). We also screened NR4A2 expression on short time upon stimulation by PDGF BB (Fig. 1D) or TGF-$\beta$ (Fig. 1E). It was revealed NR4A2 mRNA level for 12 h increased significantly compared to control group meanwhile NR4A2 protein levels elevated slightly after stimulated by PDGF BB (Fig. 1F) or TGF-$\beta$ (Fig. 1G).

Further, we isolated primary HSCs from SD rats. Primary hepatic stellate cells can be spontaneously stimulated and proliferate several days after isolation (*Davis & Vucic, 1988*). We observed that NR4A2 mRNA expression decreased significantly while $\alpha$-SMA level was increased 2–7–folds in primary hepatic stellate cells after isolation (Fig. 2).

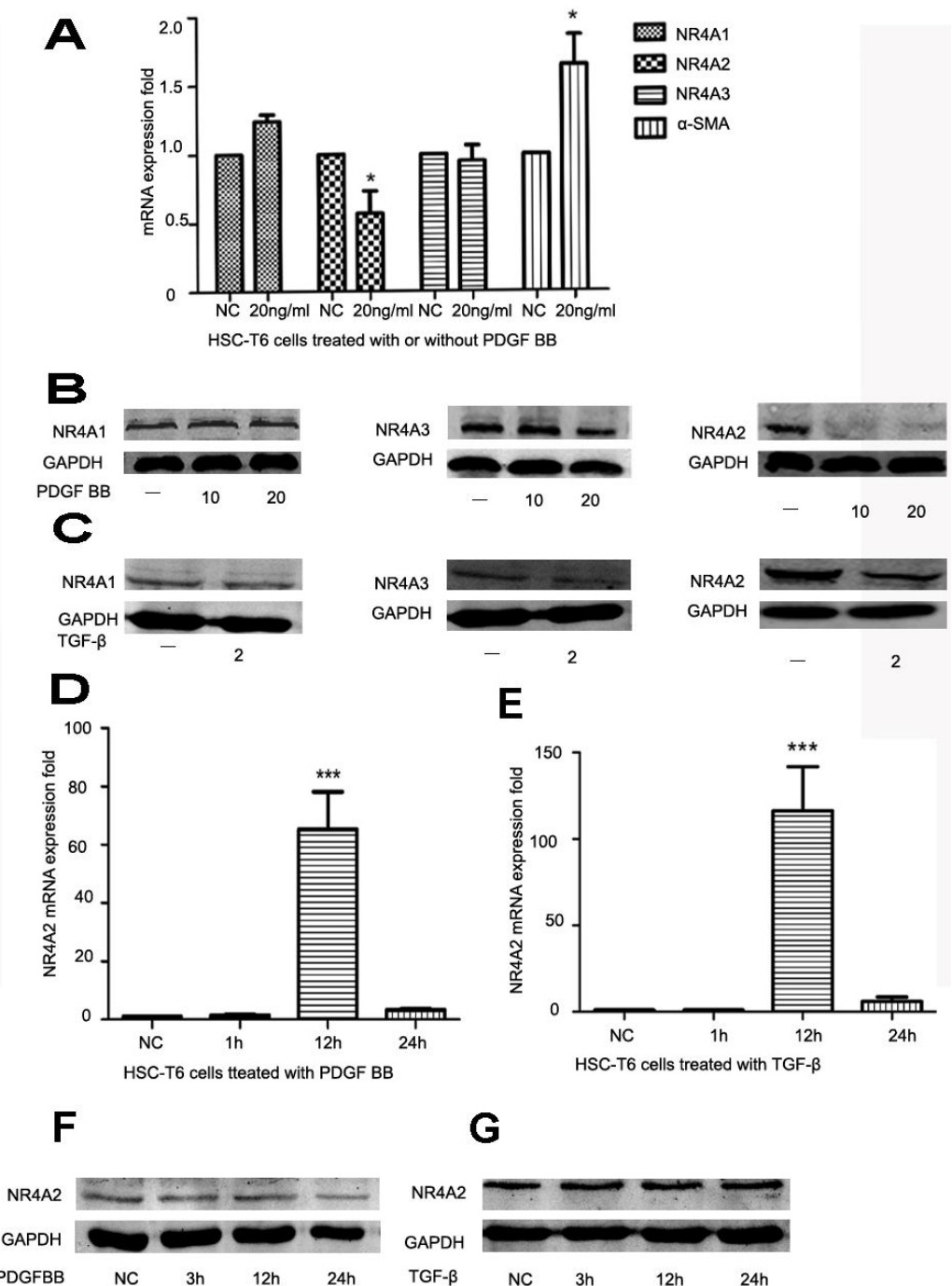

**Figure 1** **Expression of NR4A1, NR4A2, NR4A3 and α-SMA in HSC-T6 cells.** (A) Expression of NR4A1, NR4A2, NR4A3 and $\alpha$-SMA in HSC-T6 cells stimulated with or without PDGF BB (20 ng/ml) for 48 h were examined by real-time PCR ($n \geq 3$). *$P < 0.05$ vs. NC. (B) Western blot analysis of NR4A1, NR4A2 and NR4A3 in HSC-T6 cells stimulated by PDGF BB in different concentrations (0 ng/ml, 10 ng/ml and 20 ng/ml) for 48 h. (C) Western blot analysis of NR4A1, NR4A2 and NR4A3 in HSC-T6 cells stimulated with or without TGF-$\beta$ (2 ng/ml) for 48 h. 

**Figure 1 (…continued)**
(D) Real-time PCR analysis of NR4A2 levels in HS-T6 cells stimulated by PDGF BB (20 ng/ml) for 1 h, 12 h and 24 h ($n \geq 3$). ***$P < 0.001$ vs. NC. (E) Real–time PCR analysis of NR4A2 levels in HSC-T6 cells stimulated by TGF-$\beta$ (2 ng/ml) for 1 h, 12 h and 24 h ($n \geq 3$). ***$P < 0.001$ vs. NC. (F) Western blot analysis of NR4A2 levels in HSC-T6 cells stimulated by PDGF BB (20 ng/ml) for 3 h, 12 h and 24 h. (G) Western blot analysis of NR4A2 levels in HSC-T6 cells stimulated by TGF-$\beta$ (2 ng/ml) for 3 h, 12 h and 24 h.

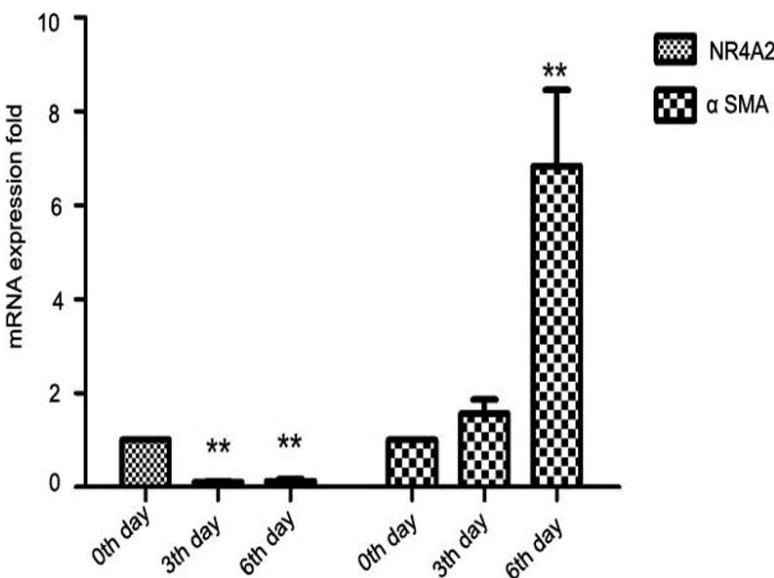

**Figure 2** **Expression of NR4A2 and $\alpha$-SMA in primary hepatic stellate cells after isolation were examined by real-time PCR ($n \geq 3$).** **$P < 0.01$ vs. 0th day group. Results are expressed as mean $\pm$ SD.

## NR4A2 expression in fibrotic liver tissue decreased

SSkip[0.25em plus .1em minus .1em] $\alpha$-SMA expressed mainly in fibrotic area where extracellular matrix accumulated whilst NR4A2 expressed in both fibrotic and non-fibrotic area of liver tissue. However, fibrotic area was dominant in fibrotic liver compared with normal liver. Immunohistochemistry analysis illustrated lower expression of NR4A2 in fibrotic area than non-fibrotic area (Figs. 3A and 3B). The same result was observed by immunofluorescence analysis (Fig. 3C).

## NR4A2 inhibited $\alpha$-SMA and Col1 expression in HSCs

To clarify the correlation between NR4A2 and liver fibrosis, we knocked down NR4A2 via three siRNAs and examined the expression of extracellular matrix markers in HSC-T6 cells. RT-PCR analysis showed that the mRNA level of NR4A2 decreased by more than 50% (Fig. 4A) while $\alpha$-SMA and Col1 increased by more than 50% (Figs. 4B and 4C).

## NR4A2 suppressed cell proliferation, modulated cell cycle and promoted apoptosis

After NR4A2 knocked down in HSC-T6 cells, we analyzed cell cycle and apoptosis by flow cytometry and examined optical density by CCK8 assay. Flow cytometry showed decreased

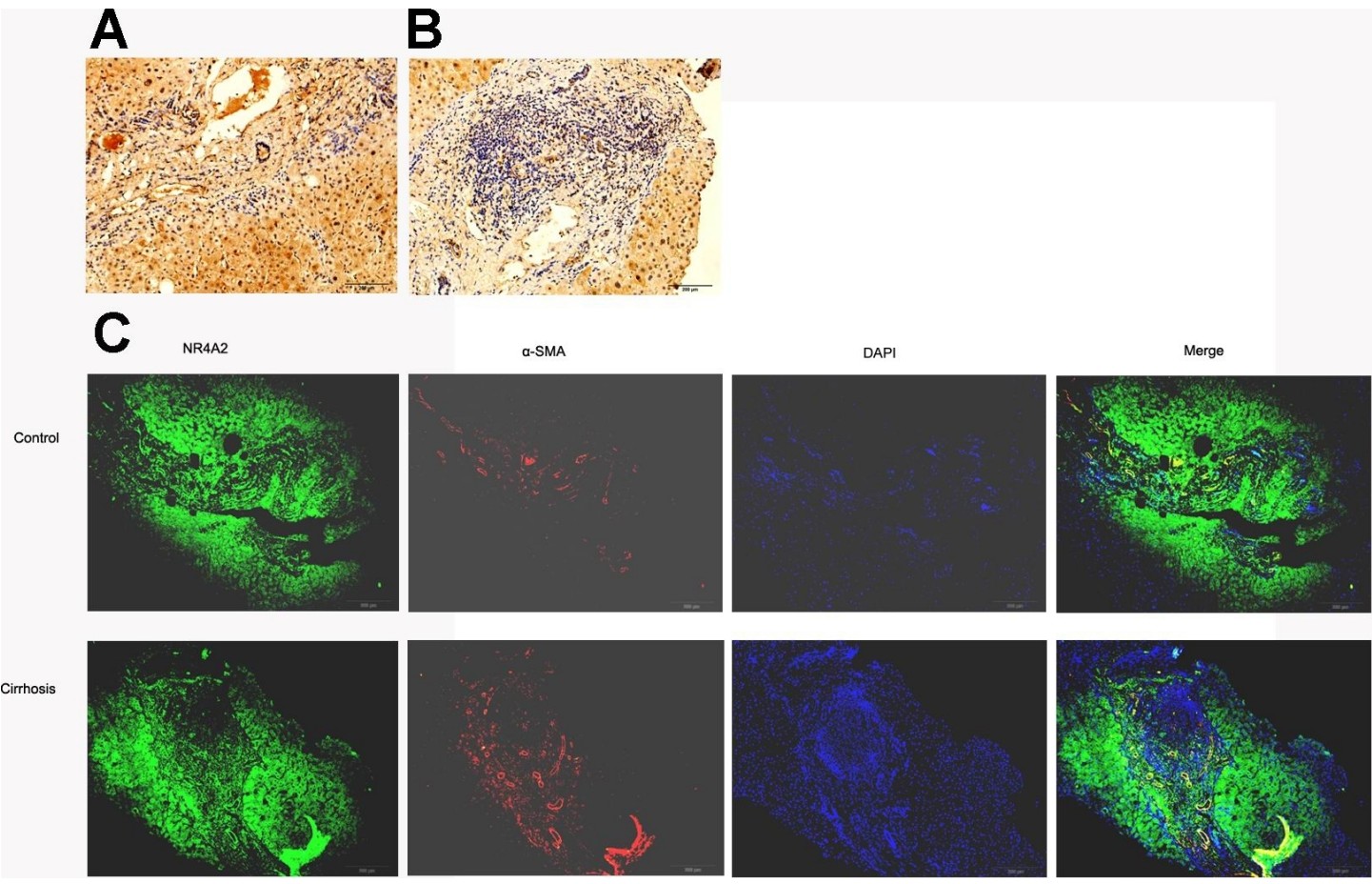

**Figure 3** **NR4A2 expression in human liver tissues.** NR4A2 expression was detected in normal liver (A) and cirrhotic liver (B) by Immunohisto-chemistry. (C) NR4A2 expression in human normal and cirrhosis liver tissue was confirmed by double staining with anti-NR4A2 (green) and anti-$\alpha$ SMA (red) antibodies. DAPI staining (blue) and the merged image are shown. *Scale bar* 200 $\mu$m.

percentage of cells in G1 phase and increased percentage in S stage compared with control group after NR4A2 knockout (Figs. 5A and 5B). NR4A2 knockout also led to reduced apoptosis (Figs. 5C and 5D) and promoted proliferation (Fig. 5E) in HSC-T6 cells.

### NR4A2 upregulation of MAPK pathway in HSC proliferation

In liver fibrogenesis, MAPK is an important signaling pathway which consists of ERK1/2, P38 and JNK. Western blotting analysis indicated reduced phosphorylation of ERK1/2, P38 and JNK with NR4A2 knockout in HSC-T6 cells (Figs. 6A–6C).

## DISCUSSION & CONCLUSIONS

Liver fibrosis is prevalent worldwide and in which HSCs play a central role (*Ueno et al., 1997*). On one hand, clearance of activated HSCs by apoptosis remained appealing for antifibrotic therapy; on the other hand, activated HSCs could also revert to quiescent

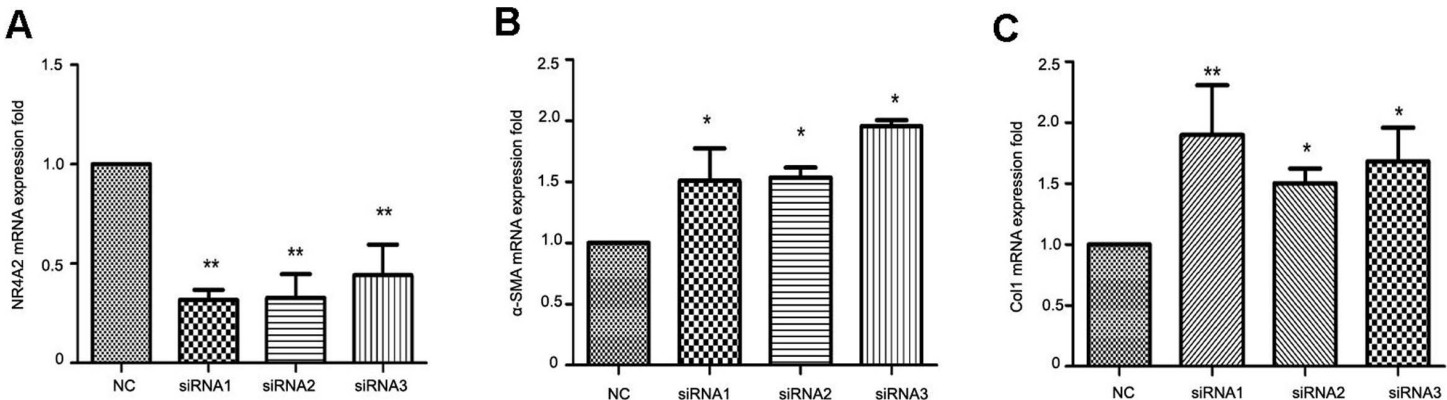

**Figure 4** **Analysis of NR4A2, α-SMA and Col1 after NR4A2 knockdown in HSC-T6 cells.** Expression of NR4A2 (A), α-SMA (B) and Col1 (C) in HSC-T6 were examined by real-time PCR ($n \geq 3$). *$P < 0.05$ vs. NC and **$P < 0.01$ vs. NC. Results are expressed as mean ± SD.

phenotype (*Olaso et al., 2001*; *Gaça et al., 2003*; *Guyot et al., 2007*). The mechanism by which activated HSCs are suppressed in liver fibrosis deserves to be extensively investigated.

Accumulated evidence suggested nuclear receptors mediate key steps in activation of HSCs (*Marra et al., 2000*; *Hellemans et al., 2004*; *Milliano & Luxon, 2005*). Peroxisome proliferator activated receptor-$\gamma$ (PPAR-$\gamma$) inhibits pro-fibrogenic genes expression in quiescent HSCs. Treatment with PPAR-$\gamma$ agonists restrains HSCs from activation and collagen production (*Galli et al., 2002*; *Marra et al., 2005*; *Chen et al., 2008*; *Yang et al., 2010*). RAR and RXR exhibit similar effect on HSCs (*Marra et al., 2000*; *Qin et al., 2008*). NR4A1 exhibited lower expression in liver fibrosis. Long-time stimulation with TGF-$\beta$ down-regulates NR4A1 level while short-time stimulation leads to increased NR4A1 in human dermal fibroblasts (*Palumbo-Zerr et al., 2015*). *Yin et al. (2013)* investigated the role of NR4A members in uterine fibroids. He found expression of NA4A1, NR4A2 and NR4A3 are remarkably lower in leiomyoma than control. The reductions in NA4A2 and NR4A3 are significantly higher than NR4A1 in human fibroid (*Yin et al., 2013*). In agreement with above results, our study indicated that NR4A2 expression decreased in liver fibrosis and PDGF BB or TGF-$\beta$ activated HSCs. At the same time no significant change was observed in NA4A1 or NR4A3. NR4A2 is mainly involving in functional maintenance of dopamine neurons and associated with nervous system diseases such as Alzheimer's disease and Parkinson's disease (*Baron et al., 2012*; *Park et al., 2012*; *Aldavert-Vera et al., 2013*; *Carloni et al., 2013*; *Moon et al., 2015*). To the best of our knowledge, this is the first report on the effect of NR4A2 in liver fibrosis. To further test the effect of short-time stimulation, we assessed NR4A2 level for less than 24 h in HSC-T6 cells after stimulation. In response to PDGF BB, NR4A2 mRNA levels elevated from 1 h to 24 h and in particular its level for 12 h was increased more than 60-fold. In contrast, NR4A2 protein levels did not increased significantly from 3 h to 24 h. The similar results were obtained upon stimulation with TGF-$\beta$. To some extent, our results are consistent with Palumbo-Zerr et al.

Primary hepatic stellate cells can be spontaneously stimulated and proliferate several days after isolation meanwhile quiet HSCs transform into activated HSCs which resembles the process *in vivo* (*Davis & Vucic, 1988*). We found sharp decrease of NR4A2 expression

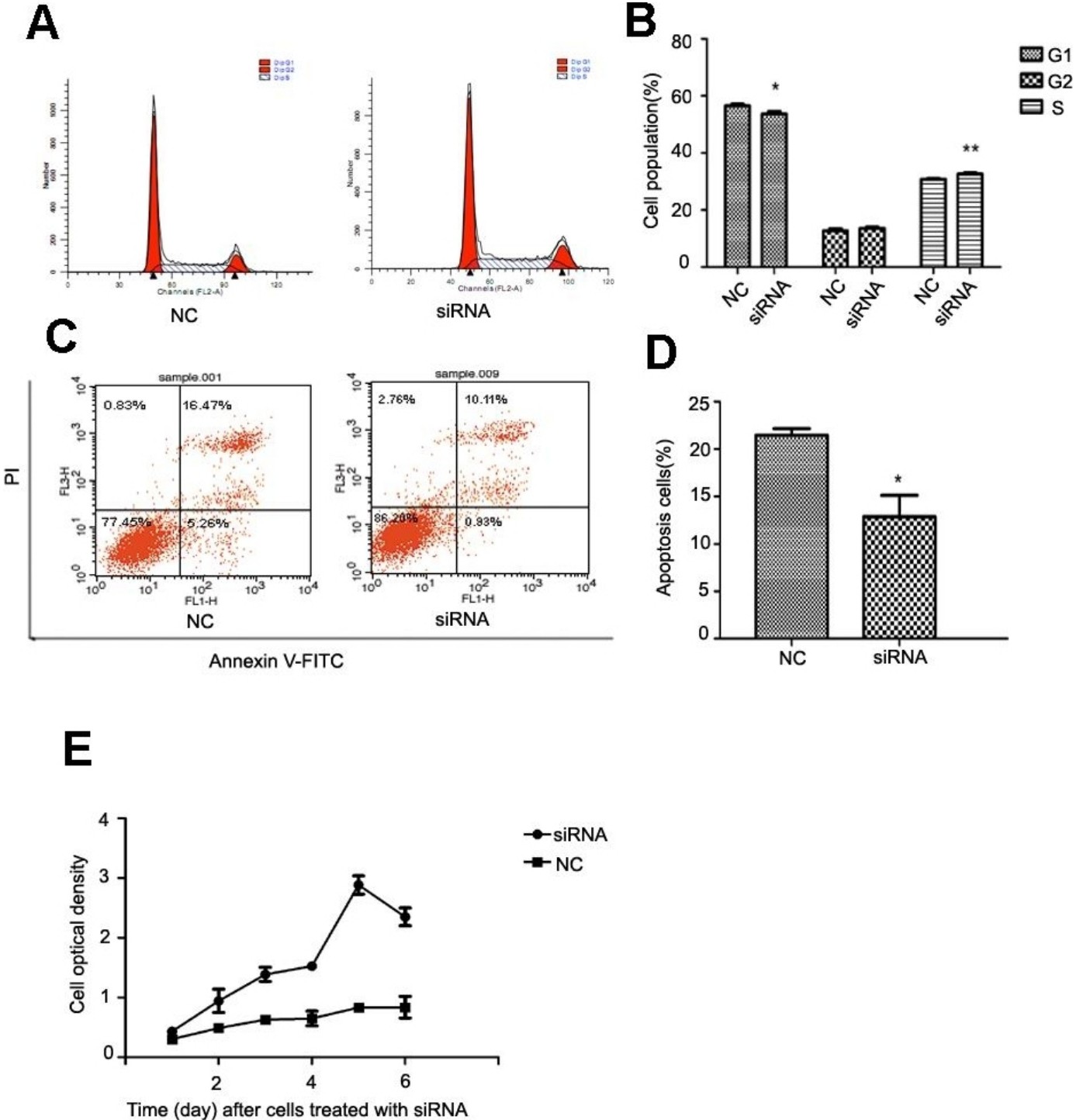

**Figure 5** **Analysis of cell proliferation, cell cycle and cell apoptosis after NR4A2 knockdown in HSC-T6 cells.** (A, B) Analysis of cell cycle. Flow cytometry graph for control and siRNA-infected HSC-T6 cells (A). Summary of cell cycle (B). The percentage of cells in G1 phase was decreased after NR4A2 knockout and the percentage of cells in S phase was increased ($n \geq 3$). $^*P < 0.05$ vs. NC and $^{**}P < 0.01$ vs. NC. (C, D) Analysis of cell apoptosis. Flow cytometry graph for control and siRNA-infected HSC-T6 cells (C). (D) Summary of cell apoptosis. The apoptosis percentage of HSC-T6 cells after NR4A2 knockdown was remarkably decreased ($n \geq 3$). $^*P < 0.05$ vs.NC. (E) Cell proliferation analyzed by cell counting kit-8 ($n \geq 3$). $^*P < 0.05$ vs NC. Results are expressed as mean ± SD.

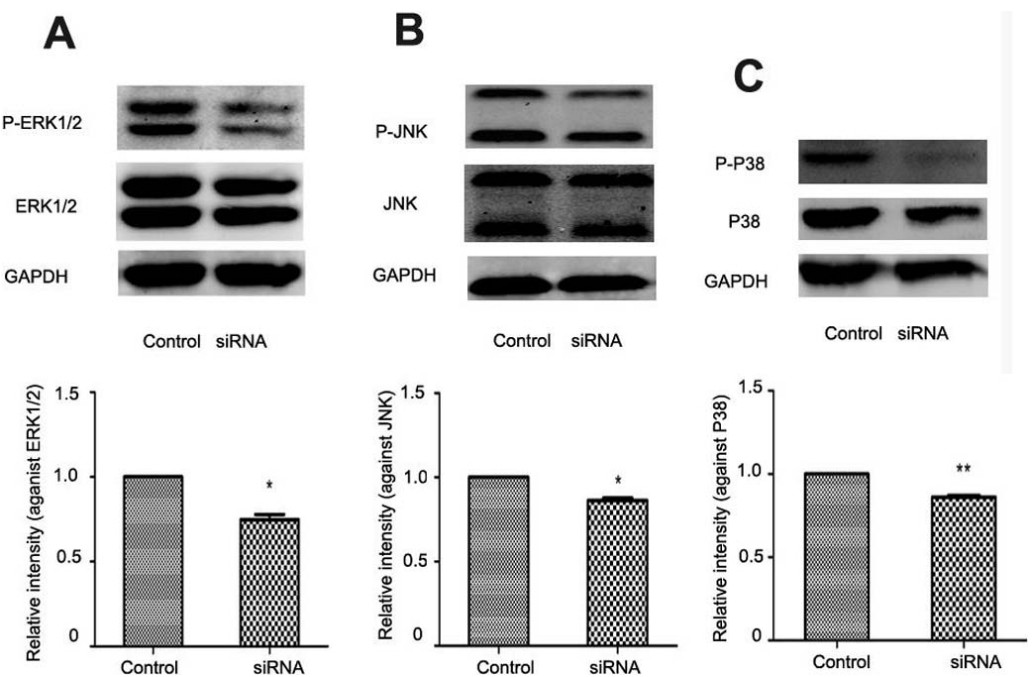

**Figure 6 Analysis of ERK1/2, P38, and JNK after NR4A2 silencing knockdown in HSC-T6 cells.** (A) Analysis of phosphorylation of ERK1/2 by Western blotting and relative intensity (P-ERK1/2 normalized against ERK1/2) ($n \geq 3$). *$P < 0.05$ vs.Con. (B) Analysis of phosphorylation of JNK by Western blotting and relative intensity (P-JNK normalized against JNK) ($n \geq 3$). *$P < 0.05$ vs. Con. (C) Analysis of phosphorylation of P38 by Western blotting and relative intensity (P-P38 normalized against P38) ($n \geq 3$). **$P < 0.01$ vs. Con. Results are expressed as mean $\pm$ SD.

and obvious increase in $\alpha$-SMA and Col1 expression in isolated hepatic stellate cells. Through NR4A2 knockout, $\alpha$-SMA and Col1, another characteristic marker for liver fibrosis down-regulated dramatically. Taken together, there is a correlation between NR4A2 and liver fibrosis and NR4A2 could down-regulate ECM genes.

NR4A2 expression in prostate tissue is conspicuously elevated and with NR4A2 knockout the cell proliferation, infiltration and migration are inhibited (*Wang et al., 2013*). Different from cancer, we found that loss of NR4A2 induced reduced apoptosis, promoted proliferation, decreased percentage of cells in G1 phase and increased percentage in S stage in liver fibrosis. On the contrary, overexpression of NR4A2 reduces cell proliferation (*Yin et al., 2013*).

NR4A2 can be mediated by a variety of signaling pathways (*Barish et al., 2005*) MAPK pathway is prominent among them. MAPK consists of ERK1/2, P38 and JNK. Our study showed that NR4A2 silencing repressed phosphorylation of ERK1/2, P38 and JNK. It may be concluded that NR4A2 suppresses cell proliferation via phosphorylating ERK1/2, P38 and JNK tin HSCs. Treatment with TGF-$\beta$ resulted in decreased NR4A2 level and knockout of NR4A2 induced reduced phosphorylation of ERK1/2, P38 and JNK. So it seems that stimulation with TGF-$\beta$ lead to suppressed phosphorylation of MAPK. This is paradoxical with the theory that TGF-$\beta$ induces MAPK/ERK activation in liver fibrogenesis (*Reimann et al., 1997*). Yet, it was confirmed that via inhibition of ERK activation TGF-$\beta$ represses

cell proliferation in pancreatic carcinoma cells and T cells (*Giehl et al., 2000*; *Luo et al., 2008*). We speculate that NR4A2 is not the sole target of TGF-$\beta$ and other genes may counteract NR4A2. Moreover, TGF-$\beta$ and NR4A2 may interact with each other. But the mechanism is unknown at present. Meanwhile, we also noticed high expression of NR4A2 in hepatocytes in liver tissue. It is not contradictory as other cells express NR4A2 *in vivo*. *Taura et al. (2010)* demonstrated that hepatocytes are not involved in liver fibrogenesis.

Our study demonstrated NR4A2 down-regulate HSCs proliferation through MAPK pathway and reduce extracellular matrix accumulation in liver fibrogenesis. However there are some drawbacks such as lack of animal experiments. The target gene of NR4A2 and the mechanism by which NR4A2 regulate HSCs need to be explored in future study. In summary, NR4A2 can inhibit proliferation of HSC through regulation of MAPK pathway and reduce extracellular matrix. NR4A2 may be a promising target for anti-fibrosis.

### Funding

This work was supported by Shanghai Committee of Science and Technology, China (Grant No.14ZR1432200) and Medical-Engineering Cross Fund Project of Shanghai Jiao Tong University (Grant No. YG2014MS21). The funders had no role in study design, data collection and analysis, decision to publish, or preparation of the manuscript.

### Grant Disclosures

The following grant information was disclosed by the authors:
Shanghai Committee of Science and Technology: 14ZR1432200.
Medical-Engineering Cross Fund Project of Shanghai Jiao Tong University: YG2014MS21.

### Competing Interests

The authors declare there are no competing interest.

### Author Contributions

- Pengguo Chen conceived and designed the experiments, performed the experiments, analyzed the data, wrote the paper, prepared figures and/or tables, reviewed drafts of the paper.
- Jie Li conceived and designed the experiments, performed the experiments, reviewed drafts of the paper.
- Yan Huo performed the experiments, prepared figures and/or tables, reviewed drafts of the paper.
- Jin Lu performed the experiments, contributed reagents/materials/analysis tools.
- Lili Wan contributed reagents/materials/analysis tools, prepared figures and/or tables.
- Bin Li and Run Gan performed the experiments.
- Cheng Guo conceived and designed the experiments, analyzed the data, prepared figures and/or tables, reviewed drafts of the paper.

## Animal Ethics

The following information was supplied relating to ethical approvals (i.e., approving body and any reference numbers):

Ethics committee of Shanghai Jiao Tong University

SYXK 2011-0128.

## Data Availability

http://dx.doi.org/10.6084/m9.figshare.1518613

http://dx.doi.org/10.6084/m9.figshare.1518612

http://dx.doi.org/10.6084/m9.figshare.1518611

http://dx.doi.org/10.6084/m9.figshare.1518597

http://dx.doi.org/10.6084/m9.figshare.1587259

http://dx.doi.org/10.6084/m9.figshare.1612158.

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
