# Peer review of "Orphan nuclear receptor NR4A2 inhibits hepatic stellate cell proliferation through MAPK pathway in liver fibrosis"

_PeerJ, doi:10.7717/peerj.1518_

## Round 0.1 · original submission · Major Revisions

I agree with both reviewers that the manuscript is poorly written and authors should rewrite the manuscript with additional attention. The quality of Western blots are very poor and I suggest to submit also full Western blot pictures for review. Both reviewers raised technical flaws in designing the experiments and they must be redesigned.

Reviewer 1 ·

Basic reporting

Chen et al. report in their manuscript “Orphan nuclear receptor NR4A2 inhibits hepatic stellate cell proliferation through MAPK pathway in liver fibrosis” that hepatic stellate cells (HSCs) play an important role in liver fibrosis. They show with different techniques that these cells express the orphan nuclear receptor NR4A2 in HSCs. Thereby NR4A2 expression was lower in fibrotic liver tissues and stimulated HSCs compared to a control group. A NR4A2 knockdown resulted in increased α-smooth muscle actin and Col1 expression. Furthermore, cell proliferation and cells being in S-phase of cell cycle increased whereas apoptosis and phosphorylation of ERK1/2, P38 and JNK decreased in stimulated cells after NR2A4 knockdown. They conclude that NR4A2 has a potential to inhibit HSC proliferation via MAPK pathway thus leading to a reduction of extracellular matrix in patients with liver fibrosis. Therefore they think that NR4A2 might be a potential target for anti-fibrotic treatment.

Most times Chen et al. write about stimulated versus unstimulated HSCs (e.g. abstract, line 22; introduction, line 60; discussion, line 263). Indeed, they should mention in the text the stimuli TGF-β and PDGF BB as this is of major importance for the whole article, a stimulus can mean anything and has to be clarified for the readership.
Figure 1B: and NR4A1 blot has an extremely poor quality due to complete overexposure and should be replaced;
Figure 1C: Western blots after stimulation with TGF-β of NR4A3 and NR4A1 should be shown.
Figure 2C: α-SMA and DAPI blots appear to be extremely dark
Figure 4F: p-JNK and p-P38 have a high background. Are better blots available?
Figure 1 B: it should be PDGF BB
Figure 1C: it should be TGF-β
Figure 3B: axis is partially cut, has to be corrected

Experimental design

Chen et al. show in Figure 1 that the NR4 receptor expression changes after 48 hours of stimulation with PDGF BB or TGF-β in HSC-T6 cells. This is a surprising long time of stimulation for me due to the following reasons: HSC-T6 cells divide in approx. 24 hours so that 48 hours stimulation time meaning approx. 2 complete cell cycles which appears to be a very long time for stimulation. Furthermore, Palumbo-Zerr et al., Nature 2015 could show that NR4A1 showed the strongest expression pattern change on RNA level within the first hour of stimulation with TGF-β whereas after 48 hours the effect was minimized. And on protein level, after 48 hours stimulation led to a comparable protein expression level whereas there was a drastic increase in the first hours (3 h to 24h) of stimulation on NR4A1 protein expression. It would be of great interest, if NR4A2 receptor expression has also such a drastic effect on stimulation with TGF-β and PDGF BB on short stimulation times like 1 h, 12 h, 24 h RNA level and something like 3 h, 12 h 24 h on protein level.

Validity of the findings

The comparison of NR4A2 with NR4A1 from data of Palumbo-Zerr et al. (lines 267-275) is important but is not sufficient as fast changes in NR4A2 expression (RNA and protein level) are not investigated and Palumbo-Zerr et al. clearly show the importance of short-term stimulation.

Results: line 209-212 and Figure 1: it is not clear what the authors mean. They took primary HSCs in culture and measured gene expression level after 3 and 6 days of culture which was obviously altered with respect to time. But the condition for this activation remains unclear. It is stated that cells are activated by the culture – what does it mean? Neither in material/methods section nor in results section there is any explanation how primary cell culture was performed, which medium, supplements were used, how often medium was changed – if at all, and if stimuli like TGF-β or PDGF BB were used.
Lines 215-219: please rewrite the whole paragraph as the statement of it remains unclear due to poorly written English.
Figure 4B: error bars are missing. Are the differences in G1 and S phase significant? If yes, please add also the significance stars. – how many times was the experiment repeated?
Figure 4C and D: correct axis labelling (Annexin versus PI) as well as the percentages of the cell distributions
Figure 4E: error bars are missing – how many times was the experiment repeated?

Additional comments

The topic of the manuscript is interesting and of relevance in the field of liver fibrosis and I think that data have a good potential to be published but it requires additional work on the manuscript, some bench work and especially a lot of work on the manuscript itself.
Manuscript has to be carefully checked for formatting as throughout the whole manuscript there are lots of errors (e.g. lines 54, 55, 56, 218, 216, 269, 284, 291).
Furthermore, it has to be carefully checked for spelling mistakes (e.g, but not limited to line 56 (studies), 142 (nm siRNA), 216 (level), 243 (participates), 264 …and – something is missing?, 281 increased, decreased.
When writing authors straight in the text to cite them, do not use first name in one case (276 Harmit S) and last name (Taura K, 290) in the other case, common rule is to give last name and no first name initials!
Material /Method section
Order numbers for all antibodies should be mentioned in material/methods section for transparence of the work.
Liver samples: did patient give their written informed consent and was sample collection in accordance to declaration of Helsinki?
PCR: please provide the details for PCR program (temperature, no of cycles) according to MIQE guidelines.
Immunofluorescence staining: what is the concentration of DAPI? (line 130)
CCK-8 assay: How long were cells incubated before CCK8 reagent was added to the cells? And how long was the siRNA transfected before cells were seeded? (lines 175-181)
References: please use one style - either with DOI number everywhere or without, but don’t mix this up.

Reviewer 2 ·

Basic reporting

The article by Chen et al. reported that NR4A2 inhibit hepatic stellate cell proliferation and thereby plays a role in MAPK pathway mediated fibrosis of liver. However, the basic writing of the manuscript is not done properly and meticulously. There are many flaws that do not fit to the standards of a good written manuscript. Authors should rewrite the manuscript in a good English and crosscheck at least once to avoid any errors. There are areas in manuscript where authors should pay attention.

Introduction:

1) Rewrite the introduction with more clarity and more explanatory rather than just reporting the prior literature in single statements. Author should pay attention that original papers are cited instead of review articles with broad overviews (for example in Line43 and 45). Also while citing in text, the author of the paper should be mentioned as “Zhang et al.” and not with full name of the first author (Line 54 and 55).

2) Line41: (MAPK)/Endoplasmic reticulum (ER); should be corrected to MAPK/ERK.

3) Line 41: Protein kinase (AKT) should be corrected to protein kinase B (AKT).

4) Line 50: References should be in chronological order.

5) The knowledge about structures and function of NR4A receptors should be introduced here and should not be discussed in the end as part of discussion (Line 258-260), if at all it is required.

6) Authors should introduce the role of other receptors of NR4A subfamily in fibrosis rather than introducing directly the NR4A2. A very important study defining the role of NR4A1 receptor in fibrosis was left out and discussed very shortly in a single line (Line 268; Palumbo-Zerr et al., 2015). Since authors are taking about NR4A receptors in liver fibrosis, this study becomes the center of their contrasting results and should be introduced and discussed extensively.

Methods:

1) Citing of materials used should be done properly with their catalog numbers. Materials purchased from a common company should be clubbed together and the company name should be mentioned at last.

2) There is no textual format for materials used, rather authors reported the material purchased in some sort of list. Please put them in a proper text format.

3) There is no need to repeat the company name after material used every time when it appears again in the rest of the manuscript.

4) Line 86: “weighting” is not a right word here, it should be “weighing”.

5) Line90: Please provide address of the place form where material was offered instead of “our hospital”.

Results:

1) In general the immunoblots are not of good quality.

2) Please formulate line 217 and 218 properly, “Where ….”.

3) Figure 4B and 4E are not having any statistical information, if experiments were performed in replicates than include the mean±S.D. The data in Figure 4B seem not to be significant and weakens the conclusion based on this result.

4) Figure 4C: Please explicatory label the X- and Y- axis with antibody and chromophore.

5) Figure captions and labeling should be uniform.

Experimental design

1) Authors should provide a good quality of immunoblots that represents their results (Figure 1B, 1C and 4F).

2) In the view of finding by Palumbo-Zerr et al., 2015 that NR4A1 is increased after TGFβ, authors should provide a quality immunoblot supporting no change in NR4A1 protein level although they observe an increase in mRNA upon PDGF stimulation of HSCs (Figure 1A and 1B)

3) The procedure to isolate hepatic stellate cells is not performed correctly. In-situ perfusion of liver and digestion with Collagenase IV/Pronase solution will lead to isolation of all hepatic cells. In order to purify stellate cells from other cells, authors should use prescribed methods such as density gradient centrifugation or FACS sorting (Mederacke et al., Nature Protocol 2015 Feb;10(2):305-15). This will avoid any artifact in the results.

4) Seeding HSCs on culture dishes lead to unphysiological activation of HSCs and overshadow the profibrogenic signals. Results derived from his type of activation should be cautiously interpreted. Authors should perform rescue experiment by overexpressing NR4A2 in primary HSCs and should show that NR4A2 expression is inhibiting the α-SMA and extracellular matrix protein mediated liver fibrosis.

5) Silencing experiments should be rescued with overexpression study in order to state the results conclusively. Without rescue experiments results should not be inferred. Often authors concluded in result sections using “Inversely, …” without supporting them with relevant experiments. (Line 232 and 239).

6) Statistical analysis should be provide where ever is necessary.

Validity of the findings

This manuscript tries to establish a relation between NR4A2 and genes involved in liver fibrosis. NR4A2 down regulation is accompanied with upregulation of α-SMA and Col1 that lead to proliferation of hepatic stellate cells. Further they tried to address that NR4A2 negatively regulates these pro-fibrotic events by inducing MAPK/ERK activation. Apart from basic and experimental problems, this manuscript has some major issues in its results that do not support its hypothesis.

1) Although the results are consistent with the observation that absence of NR4A1 (another member of NR4A subfamily) leads to fibrosis of different organs in mice, this manuscript tried to potentiate the role of NR4A2 and negate the role of NR4A1 in liver fibrosis. In my opinion, if authors want to support their hypothesis then they should examine other NR4A receptors too.
Authors have tools to examine all three NR4A receptors. Authors should also see and discuss the article by Yin et al., (Mol Endocrinol. 2013 May; 27(5): 726–740) which shows that NR4A family receptors are down-regulated in uterin fibroids. Moreover, authors should provide a quality blot for NR4A1 and explain the discrepancy with other published studies (see also point 1 in Experimental design)

2) Figure 2 C does not support the author’s observsation that NR4A2 expression decreases in fibrotic liver tissues. This statement (Line 214) is incorrect since NR4A2 staining is similar or more intense in the Cirrhosis liver sample compared to the control.

3) TGF-β is known to induce MAPK/ERK activation during liver fibrosis and also during activation of rat hepatic stellate cells (Reimann et al., FEBS Letters 403 (1997) 57-60). Since authors showed that TGF-β stimulation leads to reduction in NR4A2 level, it seems that NR4A2 reduction is inversely connected to MAPK activation. However, authors showed opposite of it and stated that NR4A2 regulates MAPK activation and silencing of it leads to inhibition of MAPK activity. Author should explain their result and discrepancies with other studies.
I estimate a no change in P-ERK, P-JNK and P-P38 phosphorylation in the view that unphosphorylated proteins also show a decrease in siRNA-treated samples. Intensity of phosphorylated bands should be normalized to the unphosphorylated bands intensity and presented in fold decrease, if any.

Annotated reviews are not available for download in order to protect the identity of reviewers who chose to remain anonymous.

---

## Round 0.2 · Minor Revisions

The manuscript has been improved, significantly. Before accpeting it for publication, I suggest authors to correct all the language and formatting mistakes.

Reviewer 1 ·

Basic reporting

Chen et al. investigated the orphan nuclear receptor NR4A2 in hepatic stellate cells and found that it inhibits proliferation through MAPK pathway in liver fibrosis. Furthermore, they investigated cell cycle and apoptosis of cells. NR4A2 expression was examined by real-time PCR, Western blotting, immunohistochemistry and immunofluorescence analysis. NR4R2 was also silenced by siRNA.
Chen et al. did a lot of efforts to improve the quality of the manuscript. Most figures have a considerable better quality now and axes are labelled properly.
But still, formatting has to be extensively corrected through the whole text as there are many mistakes on every page. Blanks are repeatedly set wrong or are missing: before brackets, blanks have to be set; after an opening bracket, text has to immediately follow without blank; after a comma, semicolon or full stop, there has to be a blank, but not before comma, semicolon or full stop. Formatting of references has to be checked also as the same problem is there also.
Neither in material/methods sections nor in the respective figures, there is mentioned that apparently the experiments were performed for N=4 times. This has to be included as this is an important information.
Figure legend for figure 1 is almost completely missing, maybe it got lost during uploading all the files, has to be included as there are a lot of subfigures (A-G) where the description has to be added.

Experimental design

no comments.

Validity of the findings

no comments.

Additional comments

Further mistakes that have to be corrected:
Page 3 line 2: has to be “diseases”
Material Methods: page 4 last line: it is “Fetal calf serum”
Page 6 line 3: Primer sequences … were “as follows”
Page 7 NR4A2 siRNA transfection: it is Briefly, “100 nmol siRNAs”
Page 8 Primary HSC isolation: 3rd line: “cannula ligated”
Page 9 “protein samples”
Page 10: cell apoptotic assay: full stop is missing at the last sentence.
Page 11: results: NR4A2 mRNA levels decreased “significantly” whereas α-SMA level increased “significantly”
Page 13/14 Discussion & Conclusion
“The mechanism by which activated HSCs are suppressed in liver fibrosis is deserved well of investigation.” – What does this mean? Something like “The mechanism by which activated HSCs are suppressed in liver fibroses deserves to be investigated.”?
Page 15 first line. Has to be “stimulation”, not “stimulated”
Page 16 “it may be concluded that NR4A2 suppresses cell proliferation”
Page 16 end of 1st paragraph “in vivo” has to be in italic.

Reviewer 2 ·

Basic reporting

The revised manuscript by Chen et al has improvement in its structure and writing after inclusion of the comments from previous version. Still there are many typing mistakes. I would advice authors to re-read manuscripts twice or thrice before they submit the manuscript.
Without listing all errors, some of them are 1) Line 38: space after full-stop "liver.We", 2) Line 46: "cellar region"

Figure 1 legend is missing. Once again, the reviewer would request authors to pay attention while writing figure legend. They must include all details, like concentrations, time of incubation and so on.

Experimental design

The authors should notice that by just playing with the contrast of immunoblots does not improve the quality. The reviewer was expecting nice clean blots which tells the story depicted by authors. Although the reviewer can notice the inclusion of some new blots, the quality of blots remain same. They are unclear and not representative of the conclusions made in the study.

In reply to point 2 of Validity of finding of previous version, authors insist that NR4A2 staining is weaker in cirrhosis liver tissues although its not visible by the immunofluorescence study. If part of the section is fibrotic then they should provide a blown-up picture of fibrotic versus non-fibrotic regions.

Validity of the findings

Although authors improved in manuscript presentation and writing as compared to its previous version, the untidy immunoblots make the validity of data dubious.

Figure 1 contain additional figures 1D, 1E and 1G showing a short-time regulation of NR4A2 during stimulation. The reviewer is puzzled to see a robust increase in mRNA level of NR4A2 without any increase in its protein content at 12hr time point. This non-corrrelation between mRNA and protein stands against their own observations in figure 1A and 1B, where a decrease in mRNA of NR4A2 is supported by a reduction in its protein expression. Authors should provide explanation for their observation.

Some of the study asked by reviewer (rebuttal point 4 and 5 of Experimental design) are already completed by authors, but they don't want to put the results in the manuscript. The opinion of this reviewer is that the inclusion of these results is necessary to validate the data and support the conclusions of the study.

Additional comments

Dear Authors,

I am happy that the manuscript got improved in its structure and presentation from its previous version. However, the untidy and poor quality of immunoblots make the conclusion of this study dubious. Kindly improve the quality of blots in Figure 1A and Figure 6 which are essential for acceptance of the manuscript.

---

## Round 0.3 · accepted · Accept

The revised manuscript is now accepted and no further revision is required.